# Prevalence and causes of blindness and vision impairment in Western Uganda: Findings from a rapid assessment of avoidable blindness (RAAB) survey

Mostafa Bondok[1], Moses Kasadhakawo[2], John Onyango[3], Oscar Turya[4], Khumbo Kalua[5]*

1 Section of Ophthalmology, Department of Surgery, Cumming School of Medicine, University of Calgary, Calgary, Canada, 2 Department of Ophthalmology, Mulago National Referral Hospital, Kampala, Uganda, 3 Department of Ophthalmology, Mbarara University of Science and Technology, Mbarara, Uganda, 4 Christian Blind Mission (CBM), Kampala, Uganda, 5 School of Population and Public Health, University of British Columbia Vancouver, Vancouver, British Columbia, Canada

* khumbo.kalua@ubc.ca

## Abstract

### Purpose

To determine the prevalence and causes of blindness and vision impairment (VI) among adults aged ≥50 years in Western Uganda.

### Methods

A population-based cross-sectional survey was conducted in Western Uganda (July-August 2023) using RAAB7. Adults aged ≥50 years who had resided in the study districts for at least six months in the past year were eligible. Participants were identified through door-to-door household visits using a two-stage cluster sampling approach. Primary outcomes include prevalence of blindness and VI and its causes. Secondary outcomes include cataract surgical coverage (CSC), effective CSC (eCSC), refractive error coverage (REC), and effective REC (eREC).

### Results

A total of 3,125 participants were examined (54.1% female). The adjusted prevalence of blindness (presenting visual acuity (PVA) <3/60) was 0.9% (95% CI: 0.5–1.3%). Severe, moderate, and mild VI were found in 0.9% (95% CI: 0.4–1.3%), 4.5% (95% CI: 3.3–5.8%), and 3.8% (95% CI: 3.0–4.6%), respectively. Untreated cataract was the leading cause of bilateral blindness (49.4%). The CSC and eCSC at the < 6/12 threshold were 19.7% and 7.3%, respectively. Only 19.4% of 108 operated eyes achieved good outcomes (PVA ≥ 6/12). The main barriers to cataract surgery included lack of awareness (32.8%), cost (23.9%), and perceived lack of need (20.9%). The

**Data availability statement:** All relevant data are within the manuscript and its Supporting Information files.

**Funding:** This research was funded by Christian Blind Mission (CBM), Uganda Office.

**Competing interests:** The authors have declared that no competing interests exist.

adjusted prevalence of uncorrected refractive error as a cause of moderate VI was 1.6% (95% CI: 1.1–2.0%), and mild VI was 2.8% (95% CI: 2.2–3.5%). REC was 1.0%, while eREC was 0.6% (95% CI: 0.0–1.4%).

## Conclusion

Blindness and vision impairment remain major public health issues in Western Uganda, primarily due to untreated cataract and uncorrected refractive error. Poor post-operative outcomes highlight the urgent need to improve surgical quality. These findings may guide targeted interventions and policy to strengthen eye care services.

## Introduction

An estimated 596 million people globally have distance vision impairment (VI), of which 43 million are blind [1]. The burden of VI and blindness is disproportionately high in low- and middle-income countries, particularly in sub-Saharan Africa, where current estimates suggest the prevalence of bilateral blindness are eight times higher than in all high-income countries [2]. The reasons for this are multifactorial [3], including a shortage of trained eye care professionals [4], a limited number of ophthalmology training programs [5], and existing barriers preventing individuals from presenting for treatment [6]. Without urgent action, the number of people affected by VI is expected to rise due to population growth, aging demographics, and urbanisation [7]. Cataract is the leading cause of blindness worldwide among adults aged ≥50 years, affecting an estimated 15.2 million people, followed by glaucoma (3.6 million) and uncorrected refractive error (2.3 million) [8]. Uncorrected refractive errors and cataracts are also the leading causes of moderate VI [9].

Recognizing the urgency of this issue, the World Health Organization (WHO) has prioritized strengthening integrated people-centered eye care services to reduce preventable blindness [2]. Resolution WHA73.4 sets global eye care targets for 2030, emphasizing effective refractive error coverage and cataract surgical coverage [10]. These initiatives build on *VISION 2020: The Right to Sight*, which encouraged countries to develop national action plans to eliminate avoidable blindness [11]. However, the lack of country-specific epidemiological data remains a major barrier to effective planning and resource allocation. To address this, WHO recommends conducting Rapid Assessments of Avoidable Blindness (RAAB) [2]. RAAB are standardized population-based surveys designed to estimate the prevalence and causes of blindness among adults aged 50 and older, who have the highest prevalence of VI [12].

Nearly 400 RAAB surveys have been conducted globally, with more than 80 conducted in sub-Saharan Africa and five in Uganda to date [13]. The most recent RAAB in Uganda was conducted in 2015 across five districts in Northern Uganda, reporting an estimated prevalence of blindness of 3.1% among 3,850 participants, with similar rates in males (3.2%) and females (3.1%) [14]. However, the growing population, changing demographics, and evolving health system capacities necessitate updated data to inform evidence-based decision-making and guide the development of a new national eye health strategy.

This study aimed to fill this gap by conducting a RAAB in Western Uganda across four districts, providing up-to-date estimates on the prevalence and causes of blindness and VI among adults aged 50 years and older. Additionally, it assessed cataract surgical coverage, post-operative outcomes, and barriers to cataract surgery uptake, as well as refractive error coverage. The findings will inform eye health policy planning and help ensure that Uganda's eye care programs are inclusive, accessible, and responsive to the needs of the population.

## Materials and methods

This cross-sectional RAAB survey was conducted between July 1 and August 8, 2023, in Western Uganda, following the standard RAAB7 methodology [15]. This study adheres to the strengthening the Reporting of observational studies in epidemiology (STROBE) guidelines [16], and the Declaration of Helsinki.

### Study setting

The study was undertaken in four districts in Western Uganda: Mbarara, Rwampara, Ibanda, and Isingiro, which had a combined population of 1,480,300, according to the Uganda Bureau of Statistics (UBOS) 2022 estimates. Data were collected by five survey teams, each consisting of an ophthalmologist and a clinical officer.

### Sample size calculation

The minimum required sample size was 4,895 participants, calculated based on an estimated prevalence of blindness of 3.1% among adults aged ≥50 years in Karamoja, Uganda, in 2015 [14]. The sample size was adjusted to achieve 20% precision, accounting for a 10% non-participation rate and a design effect of 1.5. This resulted in a total of 98 clusters, with 50 adults per cluster.

### Study population

Adults aged ≥50 years residing in the study districts for at least six months within the past year were eligible for inclusion. This study did not involve institutionalized populations or any other special groups. There was no exclusion criteria based on sociodemographic factors, language abilities, or other characteristics.

### Sampling methodology

Following RAAB7 protocol, a two-stage cluster sampling approach was used. In the first stage, clusters were selected randomly from all enumeration areas in the region, using a probability proportional to size (PPS) method. In the second stage, following cluster selection, 50 eligible participants were chosen using compact segment sampling. This methodology ensures that all individuals, including those from marginalized groups, have an equal chance of selection.

### Handling of absent participants

Eligible participants were identified through door-to-door household visits. If an eligible participant was not available, the survey team scheduled a revisit. In cases where an entire household was unresponsive or unavailable, the team attempted a return visit. Individuals who refused to participate were too ill to undergo an examination or were unable to communicate were classified as non-responders and excluded from the analysis.

### Training and quality assurance

All team members underwent pre-training via remote sessions before attending an in-person five-day training program in June 2023, facilitated by a certified RAAB trainer (KK). The training covered standard RAAB7 protocols, including study methodology, visual acuity (VA) assessment techniques, examination protocol, survey sections, data entry and

 

management, and simulation training. Inter-observer variability (IOV) testing was performed at Mulago Hospital using a sample of 50 patients. Agreement was assessed using Kappa statistics, with all teams achieving an acceptable (≥0.60 Kappa) threshold of agreement before proceeding to fieldwork. This was followed by pilot testing in a separate cluster outside the study area to refine data collection procedures under real-world conditions.

### Eye examination

Uncorrected visual acuity (UCVA) was measured in each individual eye, using a tumbling E-chart. Individuals who habitually wear spectacle correction for distance vision had their corrected visual acuity (CVA) re-measured with spectacles on. In both groups, when VA (UCVA and/or CVA) was < 6/12, the visual acuity with pinhole testing (PinVA) was performed to assess for best-corrected visual acuity (BCVA).

Presenting visual acuity (PVA) was defined as UCVA for participants without spectacles, and CVA for participants who habitual wear distance correction, as per RAAB7 protocol. Following the International Classification of Diseases (ICD-11) definitions [17], VI was categorized as:

- **Blindness**: PVA < 3/60 in the better seeing eye
- **Severe VI**: PVA ≥ 3/60 but <6/60 in the better seeing eye
- **Moderate VI**: PVA ≥ 6/60 but <6/18 in the better seeing eye
- **Mild VI**: PVA ≥ 6/18 but <6/12 in the better seeing eye

In addition to PVA, each participant underwent a structured ocular history and examination, including: (i) participant demographics (age, sex, spectacle use history); (ii) lens examination without mydriasis of all participants irrespective of PVA using a portable slit-lamp; (iv) classification of the principal cause of VI for each eye presenting with a VA < 6/12; (v) dilated fundus exam with direct ophthalmoscope and/or slit lamp when presenting VA < 6/12 and not due to cataract, corneal scar or refractive error; (vi) assessment of barriers to cataract surgery, if cataract was identified as the primary cause of VI; and (vii) cataract surgical history (age at time of surgery, place, type of surgery, cost) for previously operated individuals, including eliciting the cause of borderline or poor post-operative visual outcomes (defined as VA < 6/12), if applicable.

### Ethical considerations and data collection

All participants provided informed written consent before enrollment. Financial compensation was not provided; however, participants received treatment or referral to local eye care services as needed. Ethical approval was obtained from the Mulago Hospital Research and Ethics Committee (MHREC), and Uganda National Council for Science and Technology (UNCST).

All data were collected electronically using encrypted, personal identification number (PIN)-protected mobile devices and uploaded in real-time to the Peek Vision encrypted server, hosted in compliance with General Data Protection Regulation (GDPR) standards. Data processing and storage adhered to the RAAB7 Data Sharing Agreement.

### Results

A total of 4,895 eligible adults aged 50 years and older were identified for participation. Of these, 4,725 individuals were successfully examined, yielding a response rate of 96.5% (Table 1). The response rate was similar between males (96.3%) and females (96.7%). Among the non-responders, 1.9% were unavailable, 1.0% were incapable of participating, and 0.7% refused. The study sample comprised 2,572 (54.4%) females and 2,153 (45.6%) males. The age distribution was 47.9% aged 50–59 years, 27.9% aged 60–69 years, 15.4% aged 70–79 years, and 8.8% aged 80 + years. Compared to the census population, males in the younger age group (50–59 years) were slightly underrepresented, whereas females in the older age groups (≥70 years) were overrepresented (Supplemental Material, S1 Fig).

**Table 1. Participant characteristics.**

| | Female, No (%) | Male, No (%) | Total, No. (%) |
|---|---|---|---|
| **Examination status of enrolled participants** | | | |
| Examined | 2572 (96.7) | 2153 (96.3) | 4725 (96.5) |
| Refused | 21 (0.8) | 11 (0.5) | 32 (0.7) |
| Incapable | 22 (0.8) | 25 (1.1) | 47 (1.0) |
| Unavailable | 45 (1.7) | 46 (2.1) | 91 (1.9) |
| **Total** | 2660 (100) | 2235 (100) | 4895 (100) |
| **Age, years** | | | |
| 50–59 | 1268 (49.3) | 995 (46.2) | 2263 (47.9) |
| 60–69 | 661 (25.7) | 657 (30.5) | 1318 (27.9) |
| 70–79 | 397 (15.4) | 333 (15.5) | 730 (15.4) |
| 80+ | 246 (9.6) | 168 (7.8) | 414 (8.8) |
| **Total** | 2572 (100) | 2153 (100) | 4725 (100) |

### Prevalence of bilateral vision impairment

In the population aged 50 years and older, the age- and sex-adjusted prevalence of blindness was 0.9% (95% CI: 0.5–1.3), similar among females (0.9%, 95% CI: 0.4–1.5%) and males (0.9%, 95% CI: 0.4–1.4%) (Table 2). The prevalence of severe VI was 0.9% (95% CI: 0.4–1.3%), moderate VI was 4.5% (95% CI: 3.3–5.8%), and mild VI was 3.8% (95% CI: 3.0–4.6%) of the population.

### Causes of bilateral vision impairment and blindness

The leading principal cause of bilateral blindness was untreated cataract (49.4%), followed by age-related macular degeneration (AMD) (15.7%), and other posterior segment disease (14.5%) (Table 3). Posterior segment disease excluded AMD, diabetic retinopathy (DR), and myopic degeneration. Among individuals with moderate VI, uncorrected refractive error (25.5%) and cataract (43.1%) were the predominant causes (Table 3). Overall, 86.8% of mild and 68.6% of moderate VI cases were considered treatable (defined as uncorrected refractive error, uncorrected aphakia, or untreated cataract), as shown in Supplemental Material, S1 Table.

### Cataract surgical coverage

The age- and sex- adjusted cataract surgical coverage (CSC) at the PVA < 6/12 threshold was 14.9% overall (Table 4). However, effective cataract surgical coverage (eCSC), which represents the number of people who have been operated on and had good surgical outcomes (PVA ≥ 6/12), was 5.1%, highlighting a 65.8% relative quality gap.

### Barriers to cataract surgery

The primary barriers to cataract surgery in individuals with bilateral cataracts resulting in a PinVA < 6/60 included lack of awareness that treatment was possible (32.8%), cost (23.9%), and the feeling that surgery was not needed (20.9%) (Table 5).

### Postoperative visual outcomes

Among 108 operated eyes, only 19.4% (n = 21) achieved good visual outcomes (PVA ≥ 6/12), with 43.5% (n = 47) classified as borderline (<6/12–6/60) and 37.0% (n = 40) as poor (<6/60) visual outcomes (Supplemental Material, S2 Table). With the use of pinhole (PinVA), a total of 38.0% (n = 41) were defined as having good visual outcomes

**Table 2. Age- and sex- adjusted prevalence of blindness and vision impairment.**

| Classification* | Female, % (95% CI) | Male, % (95% CI) | Total, % (95% CI) |
|---|---|---|---|
| Blindness | 0.9 (0.4–1.5) | 0.9 (0.4–1.4) | 0.9 (0.5–1.3) |
| Severe VI | 1.0 (0.4–1.6) | 0.7 (0.1–1.4) | 0.9 (0.4–1.3) |
| Moderate VI | 5.3 (3.6–7.0) | 3.7 (2.5–5.0) | 4.5 (3.3–5.8) |
| Mild VI | 4.2 (3.0–5.3) | 3.4 (2.5–4.4) | 3.8 (3.0–4.6) |

VI: vision impairment.

*Blindness = Presenting visual acuity (PVA) <3/60 in the better seeing eye. Severe VI = PVA ≥ 3/60 but <6/60 in the better seeing eye. Moderate VI = PVA ≥ 6/60 but <6/18 in the better seeing eye. Mild VI = PVA ≥ 6/18 but <6/12 in the better seeing eye.

**Table 3. Principal cause of blindness vision impairment.**

| Principal cause | Blind, No (%) | Severe, No (%) | Moderate, No (%) | Mild, No (%) |
|---|---|---|---|---|
| Uncorrected refractive error | 1 (1.2) | 6 (6.7) | 103 (25.5) | 192 (68.6) |
| Uncorrected aphakia | 0 (0.0) | 0 (0.0) | 0 (0.0) | 0 (0.0) |
| Untreated cataract | 41(49.4) | 57 (63.3) | 174 (43.1) | 51 (18.2) |
| Cataract surgical complications | 2 (2.4) | 1 (1.1) | 7 (1.7) | 1 (0.4) |
| Trachomatous corneal opacity | 0 (0.0) | 0 (0.0) | 1 (0.2) | 0 (0.0) |
| Other corneal opacity | 5 (6.0) | 1 (1.1) | 2 (0.5) | 1 (0.4) |
| Phthisis | 0 (0.0) | 0 (0.0) | 1 (0.2) | 0 (0.0) |
| Onchocerciasis | 0 (0.0) | 0 (0.0) | 0 (0.0) | 0 (0.0) |
| Glaucoma | 5 (6.0) | 4 (4.4) | 8 (2.0) | 0 (0.0) |
| Diabetic retinopathy | 0 (0.0) | 0 (0.0) | 1 (0.2) | 0 (0.0) |
| Age-related macular degeneration | 13 (15.7) | 7 (7.8) | 41 (10.1) | 9 (3.2) |
| Other posterior segment disease | 12 (14.5) | 14 (15.6) | 65 (16.1) | 25 (8.9) |
| Myopic degeneration | 0 (0.0) | 0 (0.0) | 0 (0.0) | 0 (0.0) |
| Other globe or CNS abnormality | 4 (4.8) | 0 (0.0) | 1 (0.2) | 1 (0.4) |
| Total | 83 (100) | 90 (100) | 404 (100) | 280 (100) |

CNS: central nervous system.

*Blindness = Presenting visual acuity (PVA) <3/60 in the better seeing eye. Severe VI = PVA ≥ 3/60 but <6/60 in the better seeing eye. Moderate VI = PVA ≥ 6/60 but <6/18 in the better seeing eye. Mild VI = PVA ≥ 6/18 but <6/12 in the better seeing eye.

(Supplemental Material, S2 Table), suggesting that an additional 23.0% (20 out of 87) of those without good visual outcomes could improve with spectacle use. Two cases of blindness were the result of a cataract surgical complication (Table 3).

## Refractive error coverage

The age- and sex-adjusted prevalence and extrapolated magnitude of uncorrected refractive error as a cause of moderate VI was 1.6% (95% CI: 1.1–2.0%, N = 2,160) and mild VI was 2.8% (95% CI: 2.2–3.5%, N = 3,903) (Table 6). Refractive error coverage (REC) was 1.0% overall, while effective refractive error coverage (eREC) was only 0.6% (95% CI: 0.0–1.4%), reflecting a significant gap in refractive services (Supplemental Material, S3 Table). This differed by gender, with males having higher REC (2.2% [95% CI: 0.0–0.0%] vs 0.0% [95% CI: 0.0–0.0%]) and eREC (1.3% [95% CI: 0.0–0.0%] vs 0.0% [95% CI: 0.0–0.0%]) compared to females. Distance vision spectacles were only used by seven (0.1%) participants, and near vision spectacles by 276 (5.8%) of participants.

**Table 4. Age- and sex- adjusted cataract surgical coverage and effective cataract surgical coverage.**

| | Female, % (95% CI) | Male, % (95% CI) | Total, % (95% CI) | Quality Gap |
|---|---|---|---|---|
| **Cataract surgical threshold <6/12** | | | | |
| CSC | 17.7 (12.5–23.0) | 10.8 (5.4–16.2) | 14.9 (11.1–18.7) | 65.8% |
| eCSC | 7.8 (4.8–10.9) | 1.2 (0–3.5) | 5.1 (3.0–7.3) | |
| **Cataract surgical threshold <6/18** | | | | |
| CSC | 21.9 (15.1–28.7) | 16.6 (8.9–24.4) | 19.8 (14.8–24.8) | 68.2% |
| eCSC | 9.2 (5.3–13.1) | 1.9 (0–5.4) | 6.3 (3.4–9.2) | |
| **Cataract surgical threshold <6/60** | | | | |
| CSC | 43.8 (29.6–58.0) | 37.8 (22.8–52.7) | 41.2 (31.1–51.2) | 64.6% |
| eCSC | 22.6 (13.7–31.5) | 4.2 (0–11.9) | 14.6 (8.4–20.8) | |
| **Cataract surgical threshold <3/60** | | | | |
| CSC | 67.1 (49.5–84.7) | 50.8 (33.1–68.5) | 59.3 (46.9–71.6) | 63.2% |
| eCSC | 36.5 (22.7–50.4) | 5.8 (0–16.1) | 21.8 (12.9–30.7) | |

CSC: cataract surgical coverage; eCSC: effective cataract surgical coverage.

**Table 5. Barriers to cataract surgery among participants with bilateral cataract and PinVA<6/60.**

| Barrier | Female, No (%) | Male, No (%) | Total, No (%) |
|---|---|---|---|
| Unaware treatment possible | 15 (31.9) | 7 (35) | 22 (32.8) |
| Surgery denied by provider | 0 (0.0) | 1 (5.0) | 1 (1.5) |
| Cannot access surgery | 2 (4.3) | 2 (10.0) | 4 (6.0) |
| Cost | 10 (21.3) | 6 (30.0) | 16 (23.9) |
| Felt not needed | 11 (23.4) | 3 (15.0) | 14 (20.9) |
| Fear | 9 (19.1) | 1 (5.0) | 10 (14.9) |
| Other | 0 (0.0) | 0 (0.0) | 0 (0.0) |
| **Total** | 47 (100) | 20 (100) | 67 (100) |

PinVA: Pinhole visual acuity

## Discussion

The adjusted prevalence of blindness in this study was 0.9%, while that of severe VI was also at 0.9%. While no previous RAAB has been conducted in this specific region of Western Uganda, comparisons can be made with the 2013 RAAB in Western Uganda's Hoima District, where the adjusted prevalence of blindness and severe VI was 1.9% and 1.8%, respectively [18]. Although the study areas do not overlap, both regions have historically similar poverty rates [19], suggesting a potential improvement in eye health outcomes. In contrast, Karamoja, a region in Northern Uganda with one of the highest national poverty rates [19], reported a 2023 prevalence of blindness and severe VI of 4.9% and 3.1%, respectively [14], highlighting significant regional differences likely influenced by economic factors.

Untreated cataract remains the leading cause of blindness, accounting for 49.4% of cases in our study. Unlike in 2013, when a significantly higher proportion of blindness from untreated cataract was reported in women (CSC<3/60: 70.9% in men vs. 37.4% in women) [18], our study found more comparable rates between sexes (CSC<3/60: 50.8% in men vs. 67.1% in women).While this suggests an improvement in surgical coverage, the difference was not statistically significant. This may reflect improvements in equity of access, resulting from strategies designed to enhance women's access to cataract surgery, including community engagement, service delivery adaptations, and gender-responsive planning [20]. Community approaches included working with women's groups, community health workers, and household decision-makers

**Table 6. Age- and sex- adjusted prevalence of blindness and vision impairment due to refractive error.**

| Level of VI | Female | | Male | | Total | |
|---|---|---|---|---|---|---|
| | % (95% CI) | N* | % (95% CI) | N* | % (95% CI) | N* |
| Blindness | 0.0 (0.0–0.1) | 3 | 0.0 (0.0–0.0) | 0 | 0.0 (0.0–0.0) | 3 |
| Severe | 0.2 (0.0–0.3) | 106 | 0.0 (0.0–0.1) | 3 | 0.1 (0.0–0.2) | 108 |
| Moderate | 1.7 (1.0–2.3) | 1161 | 1.4 (0.9–2.0) | 999 | 1.6 (1.1–2.0) | 2160 |
| Mild | 3.1 (2.1–4.0) | 2152 | 2.5 (1.7–3.3) | 1750 | 2.8 (2.2–3.5) | 3903 |

VI: vision impairment.

*N represents the extrapolated magnitude within the population.

Blindness = Presenting visual acuity (PVA) <3/60 in the better seeing eye. Severe VI = PVA ≥ 3/60 but <6/60 in the better seeing eye. Moderate VI = PVA ≥ 6/60 but <6/18 in the better seeing eye. Mild VI = PVA ≥ 6/18 but <6/12 in the better seeing eye.

to raise awareness and mobilize women, while outreach and surgical services were adapted through women-only camps, priority management for pregnant or breastfeeding women, gender-segregated waiting areas, same-day surgery, and transport or financial support to reduce indirect barriers. Programmes also used sex-disaggregated baseline data and gender-specific targets to inform planning, complemented by capacity building for health workers and ongoing monitoring of gender-disaggregated outcomes. Systematic case finding, school-based health education, and documentation of best practices further supported equitable service delivery, addressing both practical and societal barriers to surgical uptake among women [20]. Ongoing efforts are needed to ensure all populations, particularly women and older adults, can access high-quality cataract services.

Notably, eCSC reflects the results of all surgeries received by participants, many of which were decades earlier using superseded techniques [21]. Post-operative visual outcomes were borderline in 43.5% and poor in 37.0% of cases. It is important to consider that in the context of RAAB, these VA measurements may come years to decades after surgery. Poor outcomes may be the result of pre-existing conditions (poor selection), or long-term sequelae after surgery (progressive ocular comorbidities, such as retinal diseases), rather than surgical factors alone [21]. A study of nearly 2000 patients undergoing cataract surgery in Kenya found that half of patients with poor VA at two-month follow-up could be attributed to pre-existing retinal disease [22].

Despite these considerations, the cataract quality gap in this study remains substantial at 64.6%. Being well over 25% threshold, this emphasizes that improving surgical quality should take precedence over simply increasing surgical volume [21], as enhanced quality directly impacts functional outcomes. Several strategies, identified in other settings, could enhance cataract surgery outcomes in Western Uganda, though their effectiveness in this population remains uncertain.

First, structured post-surgical monitoring is strongly associated with better results [22]. In settings with poor follow-up, early VA assessment within three days of surgery, and unprompted assessments of patients returning after 6 weeks are considered valid measures of operative quality [23]. This monitoring should be conducted at the individual surgeon or center level, not for direct comparisons, but to identify areas for improvement [24], including better selection of surgical candidates. The extent of follow-up among participants in our study remains unclear, as this was not assessed.

Second, investments in optical biometry and enhanced surgical training could optimize outcomes and minimize deviations from target refraction [25,26]. Finally, human resource constraints remain a significant challenge, with the most recent data revealing one ophthalmologist per 1.2 million [27], and two ophthalmology residency training programs in Uganda [5]. Beyond increased ophthalmologists, optometrist, and allied ophthalmic personnel, there remains a need for enhanced surgical training [28].

Notably, post-operative visual outcomes did not significantly differ by surgical setting or sex, marking a contrast from previous studies that reported gender disparities in surgical access and outcomes [6,18,21,29]. Prior research indicates

men are more likely to undergo cataract surgery due to gendered social roles and greater access to household resources [6,21,29].

The main barriers to cataract surgery in our study were lack of awareness that treatment was available (32.8%), cost (23.9%), and fear of surgery (14.9%). These findings are consistent with previous RAABs in Uganda [14,18], highlighting the persistent need for public health education to promote cataract surgery uptake [30]. A study at Mulago National Referral Hospital in Kampala, Uganda, found that financial constraints, the perception that surgery was unnecessary, and the ability to see with one eye were the primary reasons for delayed cataract surgery among operable patients [31]. Several factors have been associated with higher cataract surgery uptake [6], including male sex [32,33], relatively younger age (50–60 years vs. > 70 years) [33], and supportive family attitudes toward surgery [34], although other studies have not found the same significant associations [35–38]. Interventions shown to improve uptake include eliminating direct surgical costs [32]; however, even when surgery is free, uptake may remain lower than expected [33]. This may be due to indirect cost such as transportation, and lost income from time off work [39,40]. Regular outreach programs have also been reported to increase uptake [37,41], though it is unclear whether this effect reflects reduced travel barriers or greater trust in familiar service delivery models [6]. Further research is needed to move beyond self-reported barriers and evaluate interventions in real-world clinical settings to determine which strategies effectively improve cataract surgery uptake.

The burden of uncorrected refractive error in this study highlights ongoing gaps in refractive services. While uncorrected refractive error was a leading cause of moderate and mild visual impairment, REC and eREC remained critically low at 1.0% and 0.6%, respectively. These findings are consistent with previous RAAB conducted in Uganda and other sub-Saharan African countries, where refractive services remain underdeveloped [42]. Given that uncorrected refractive error is easily correctable with spectacles, these low coverage rates are particularly concerning. Unless addressed promptly through coordinated global action, such as the WHO's SPECS 2030 initiative to strengthen refractive error care [43], countries like Uganda risk falling behind and failing to achieve global targets.

Limited availability and affordability of refractive services likely contribute to this gap. In our study, only 0.1% of participants used distance vision spectacles, and just 5.8% used near vision spectacles, reinforcing evidence that spectacles remain largely inaccessible. Prior studies in Uganda have similarly identified cost as a major barrier to spectacle uptake [44]. Additionally, there are only 10 optometrists in Uganda [27], further exacerbates human resource shortages that limit access to affordable refractive services.

### Limitations

This study has several limitations. RAAB surveys focus on assessment of adults aged 50 years and older, using results as a proxy for the total population [45]. Furthermore, RAAB assigns a principal cause of VI per participant, such that ocular multi-morbidity is not captured. The cross-sectional nature of the study does not allow for the assessment of trends over time. While the response rate was high (96.5%), younger males (50–59 years) were underrepresented, whereas females in older age groups (≥70 years) were overrepresented, necessitating the use of age- and sex-adjusted prevalence estimates. This was likely the case given males are more likely to be at work when survey teams visit. The RAAB methodology, which prioritizes quantifying causes of vision loss most amenable to public health interventions [15], may underestimate posterior segment diseases like glaucoma and diabetic retinopathy as a dilated fundus exam is only performed when the VI could not be sufficiently explained by the presence of cataract, corneal scar or refractive error [46]. Despite these limitations, the study provides critical data to guide eye health policy and resource allocation in Uganda.

### Conclusions

This study provides updated estimates on the prevalence and causes of blindness and vision impairment in Western Uganda. Untreated cataract remains the leading cause of blindness, with low effective cataract surgical coverage (eCSC), reflecting poor surgical outcomes. Uncorrected refractive error significantly contributes to moderate and mild vision

impairment. Strengthening cataract surgical quality and expanding refractive services are critical. These findings may guide national eye health policy to improve access and outcomes for all.

## Supporting information

**S1 Fig. Comparison of survey population and census population by age and sex.**
(DOCX)

**S1 Table. Principal cause of blindness and vision impairment by intervention category.**
(DOCX)

**S2 Table. Post-operative visual outcomes after cataract surgery.**
(DOCX)

**S3 Table. Age- and sex- adjusted distance refractive error coverage and effective refractive error coverage.**
(DOCX)

## Acknowledgments

The authors would like to thank all who contributed to the success of this study, including the data collectors in Uganda and the Rapid Assessment of Avoidable Blindness (RAAB) team at the London School of Hygiene & Tropical Medicine for their support with data analysis.

## Author contributions

**Conceptualization:** Mostafa Bondok, Moses Kasadhakawo, John Onyango, Oscar Turya, Khumbo Kalua.

**Data curation:** Moses Kasadhakawo, John Onyango, Oscar Turya, Khumbo Kalua.

**Formal analysis:** Mostafa Bondok, Moses Kasadhakawo, John Onyango, Oscar Turya, Khumbo Kalua.

**Funding acquisition:** Moses Kasadhakawo, Oscar Turya, Khumbo Kalua.

**Investigation:** Mostafa Bondok, Moses Kasadhakawo, John Onyango, Oscar Turya, Khumbo Kalua.

**Methodology:** Mostafa Bondok, Moses Kasadhakawo, John Onyango, Oscar Turya, Khumbo Kalua.

**Project administration:** Moses Kasadhakawo, John Onyango, Oscar Turya, Khumbo Kalua.

**Resources:** Moses Kasadhakawo, John Onyango, Khumbo Kalua.

**Software:** Khumbo Kalua.

**Supervision:** Moses Kasadhakawo, John Onyango, Khumbo Kalua.

**Validation:** Mostafa Bondok, Moses Kasadhakawo, John Onyango, Khumbo Kalua.

**Visualization:** Mostafa Bondok, Khumbo Kalua.

**Writing – original draft:** Mostafa Bondok, Moses Kasadhakawo, John Onyango, Oscar Turya, Khumbo Kalua.

**Writing – review & editing:** Mostafa Bondok, Moses Kasadhakawo, John Onyango, Oscar Turya, Khumbo Kalua.

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
