## [Decision Letter · Decision Letter 0]

25 Aug 2025

Dear Dr. Kalua,

Thank you for submitting your manuscript to PLOS ONE. After careful consideration, we feel that it has merit but does not fully meet PLOS ONE’s publication criteria as it currently stands. Therefore, we invite you to submit a revised version of the manuscript that addresses the points raised during the review process.

We look forward to receiving your revised manuscript.

Kind regards,

Van Charles Lansingh, MD, PhD

Academic Editor

PLOS ONE

Journal Requirements:

“This research was funded by Christian Blind Mission (CBM), Uganda Office.”

3. We note that your Data Availability Statement is currently as follows: [Add Data Availability statement here]

Additional Editor Comments (if provided):

Dear Authors,

Congratulations on a very well written paper. It is specially important to generate the evidence and base line data in a region where there is a paucity of recent population based studies conducted. It will help the country to determine the steps needed to fulfill the targets set out by the WHA resolutions.

Reviewers' comments:

Reviewer's Responses to Questions

**Comments to the Author**

1. Is the manuscript technically sound, and do the data support the conclusions?

Reviewer #1: Yes

Reviewer #2: Yes

2. Has the statistical analysis been performed appropriately and rigorously?

Reviewer #1: Yes

Reviewer #2: Yes

3. Have the authors made all data underlying the findings in their manuscript fully available?

Reviewer #1: Yes

Reviewer #2: Yes

4. Is the manuscript presented in an intelligible fashion and written in standard English?

Reviewer #1: Yes

Reviewer #2: Yes

Reviewer #1: I think this is a very well set up study. I really appreciate the hard field work the ophthalmologists have done in their house to house visitis to collect this data. They adhered well to the RAAB study set up.

I would consider a small rewrite of the discussion. It is stating arguments on the reasons for an ineffective cataract surgical coverage and is stating that there were no reviews of surgeries done within 3 days and no proper biometry which might have lead to poor refractive outomces. This are however suggestions and these data were not collected in questionaires on the patients who underwent cataractsurgery. Nor was refraction done on those patients. So I suggest to keep those arguements out of the discussion. Or rewrite them as possible reasons as shown in other studies, with refferences

Reviewer #2: General observation: "It is great to see evidence being generated from the African region, where data remain limited for planning. Truly appreciate the efforts of the research team.

Overall, the article is presented well. Areas for improvement include: providing clarity on the eye examination process; avoiding repetition of definitions to enhance readability; and strengthening the Discussion section. "

ABSTRACT:

Methods: Participants, sampling, recruitment details should be included.

RAAB version, whether 6 or 7, may be mentioned.

Results: Add total participants and sex details.

CSC and eCSC should be reported at 6/12 threshold, as this is readily available in the RAAB7 version. Additional thresholds may be reported in the main text as appropriate. Reporting at 6/12 threshold will further support the conclusion in the abstract.

eREC should be defined either in methods or results section.

Keywords: eCSC, eREC may be added.

INTRODUCTION:

Line no: 83 "a shortage of trained eye care providers':-

The provider word can be replaced with 'professionals' if you refer to eye care workforce. If you are referring to eye hospital, then 'trained' word might be removed. "

Line no: 86 -88 "The leading cause of blindness globally is cataract, with an estimated 17.01 million people, followed by uncorrected refractive error with an estimated 3.70 million people':-

Please recheck the burden count in the article you cited.

Also mention the age group."

MATERIALS AND METHODS:

Line no: 150 Eye examination would be an appropriate word instead of the subheading 'study procedure'.

Line no: 150-154 "Uncorrected visual acuity (UCVA) was measured with both eyes open using a tumbling E-chart.'

As per RAAB protocol, visual acuity is measured for one eye first (while covering the other eye), followed by next eye. Kindly clarify.

'In both groups, when VA was <6/12, the 153 visual acuity with pinhole testing (PinVA) was performed to assess for best-corrected visual acuity

(BCVA).'

For pinhole criteria, specify whether it is based on UCVA/PVA"

Line no: 155-156 Check for grammatical accuracy.

Line no: 163 "In RAAB7 version, 'name' is not collected for all individuals, but only for the unavailable persons. However, these names were saved only in local tablets during data collection, not saved in RAAB database. Hence this can be removed from demography.

'optional GPS coordinates':- You may mention this only if you used this variable. Otherwise, not necessary to mention. "

Line no: 175 "and London School of Hygiene & Tropical Medicine Observational Ethics Committee'

Individual RAAB survey do not require ethical approval from LSHTM; the research team only makes an agreement for use of RAAB7 software. Therefore, this statement should be not incuded under ethics section. "

RESULTS:

Line no: 182-191 Table 1 is cited in line no 184. It is assumed that the subsequent sentences refer to the same. So there is no need to repeat citing table no. for each sentence.

Line no: 195 -200 Definition of blindness and VI is provided in the Methods section and in the accompanying table. Repetition may be avoided where possible.

Line no: 222-226 cataract surgical coverage is highlighted at 6/60 threshold, rather than the meaningful 6/12 threshold, which is essential and recommened by WHO as the benchmark that countries should aim to achieve. Moreover, you have already discussed the 2030 WHO targets in the Introduction.

Line no: 228 Quality gap can be inlcuded in Table 4

Line no: 235 "Total 'n' counts can be given in Table 5

Total counts may be provided in all tables in the last row, where possible."

Line no: 245-247 Total refractive error prevalence can be mentioned, though breakdown of VI was given.

DISCUSSION:

Nearly half of the Discussion section is repetition of results. There is scope for strengthening this section by focusing on interpretation, implications, recommendations to eye care providers, government, NGOs.

Line no: 266-268 The comparison statement seems to be incorrect. Please be careful while quoting other studies. The following is the prevalence % from the article you cited.....'Prevalence of visual impairment was found to have decreased from 6.0% (95%CI 5.2-6.9%) in 2015 to 4.9% (95%CI 4.0-5.9%]) in 2023'

Line no: 270-272 "Unlike in 2013, 271 where a significantly higher proportion of blindness from untreated cataract was observed in females (62.9% vs. 32.1%), our study found similar rates between sexes (47.9% vs. 51.4%).'

You can say that this difference was due to various initiatives of cataract programs....list programs."

Line no: 273 Mention visual acuity threshold for 2013 RAAB

Line no: 297-299 "Compared to the 2013 RAAB, where cataract surgeries were predominantly performed in government hospitals [18], our study found a nearly equal distribution of procedures across government, non-governmental organization (NGO), and private hospitals.'

In Results section, there is no statement regarding the place of surgery, therefore, it should not discussed in the Discussion section. "

**Do you want your identity to be public for this peer review?** For information about this choice, including consent withdrawal, please see our Privacy Policy

Reviewer #1: **Yes: ** Tim Westland

Reviewer #2: No

---

## [Author Response · Author response to Decision Letter 1]

24 Sep 2025

Thank you for the opportunity to revise our manuscript “Prevalence and causes of blindness and vision impairment in Western Uganda: Findings from a rapid assessment of avoidable blindness (RAAB) survey”. Below, we present our responses to reviewer comments in blue, along with the corresponding line/page numbers for any manuscript changes. “Please note that the line numbers refer to the clean version of the revised manuscript (not the tracked-changes version).

---

## [Editor Report · Decision Letter 1]

29 Sep 2025

Prevalence and causes of blindness and vision impairment in Western Uganda: Findings from a rapid assessment of avoidable blindness (RAAB) survey

PONE-D-25-28213R1

Dear Dr. Kalua,

We’re pleased to inform you that your manuscript has been judged scientifically suitable for publication and will be formally accepted for publication once it meets all outstanding technical requirements.

Kind regards,

Van Charles Lansingh, MD, PhD

Academic Editor

PLOS ONE
---

## [Editor Report · Acceptance letter]

PONE-D-25-28213R1

PLOS ONE

Dear Dr. Kalua,

I'm pleased to inform you that your manuscript has been deemed suitable for publication in PLOS ONE. Congratulations! Your manuscript is now being handed over to our production team.

Kind regards,

on behalf of

Dr. Van Charles Lansingh

Academic Editor

PLOS ONE